# Implications of monocular vision for racing drivers

**Julien Adrian**[1]*, **Johan Le Brun**[1], **Neil R. Miller**[2], **José-Alain Sahel**[3,4,5,6], **Gérard Saillant**[7], **Bahram Bodaghi**[8]

**1** Streetlab, institut de la vision, Paris, France, **2** Wilmer Eye Institute, the Johns Hopkins University School of Medicine, Baltimore, Maryland, United States of America, **3** Centre Hospitalier National d'Ophtalmologie des Quinze-Vingts, INSERM-DHOS CIC 503, Paris, France, **4** Rothschild Ophthalmology Foundation Hospital, Paris, France, **5** Sorbonne Universités, INSERM, CNRS, Institut de la Vision, Paris, France, **6** Department of Ophthalmology, University of Pittsburgh School of Medicine, University of Pittsburgh Medical Center, Pittsburgh, Pennsylvania, United States of America, **7** Fia, Paris, France, **8** Department of Ophthalmology, DHU Vision and Handicaps, Hôpital Pitié-Salpêtrière, Paris, France

* julien.adrian@streetlab-vision.com

**Data Availability Statement:** All relevant data are within the paper and its Supporting Information files.

**Funding:** This work was supported by FIA Institute, Streetlab. This research did not receive any specific

## Abstract

We performed two experiments to investigate how monocular vision and a monocular generalized reduction in vision (MRV) impact driving performance during racing. A total of 75 visually normal students or professional racing drivers, were recruited for the two experiments. Driving performance was evaluated under three visual conditions: normal vision, simulated monocularity and simulated monocular reduction in vision. During the driving scenario, the drivers had to detect and react to the sudden intrusion of an opponent's racing car into their trajectory when entering a turn. Generalized Linear Mixed Models (GLMMs) and ANOVA were then used to explore how monocular vision and monocular reduction in vision affect drivers' performance (crash and reaction time) while confronting them with critical situations. The results show that drivers under monocular condition are from 2.1 (95% CI 1.11–4.11, p = .024) to 6.5 (95% CI 3.91–11.13; p = .0001) times more likely to collide with target vehicles compared with their baseline (binocular) condition, depending on the driving situation. Furthermore, there was an average increase in reaction time from 64 ms (p = .029) to 126 ms (p = .015) under monocular condition, depending on the critical driving situation configuration. This study objectively demonstrates that monocularity has a significant impact on driving performance and safety during car racing, whereas performance under monocular reduction in vision conditions is less affected.

## Introduction

Motor racing is a particularly dangerous sport that requires being in perfect physical condition, having fast reaction times and having good vision. During car racing, where the velocities are very high, the driver's vision is essential to drive a powerful car safely and to detect and react in short timeframes to sudden and unexpected events. Yet, to date, there is a lack of literature for determining the visual requirements of a racing driver.

grant from funding agencies in the public, commercial, or not-for-profit sectors.

**Competing interests:** I have read the journal's policy and some authors of this manuscript have the following competing interests: José-Alain Sahel: Funding Sources: LabEx LIFESENSES (ANR-10-LABX-65), ERC Synergy "HELMHOLTZ" (ERC Grant Agreement #610110), Banque publique d'Investissement (Sightagain BPI-2014-PRSP-15), University-Hospital Institute "FOReSIGHT ((ANR-18-IAHU-01)", Foundation Fighting Blindness (C-CL-0912-0600-INSERM01; C-GE-0912-0601-INSERM02). Consultant: Pixium Vision; GenSight Biologics; SparingVision. Personal Financial Interests: GenSight Biologics, Prophesee, Chronolife, Pixium Vision, Tilak Healthcare, Sparing Vision. Bahram Bodaghi is consultant for the Medical Commission of the FIA. Neil R. Miller is study director for the QRK207 Clinical Treatment Trial for acute NAION (Quark Pharmaceuticals) and consultant to the Regenera pharmaceutical company. Gérard Saillant is President of the Medical Commission of the FIA. The other authors do not have any conflicts of interest to disclose. This does not alter our adherence to PLOS ONE policies on sharing data and materials.

Monocular vision is an impairment that intuitively would not allow someone to perform motor racing. Although it has been reported that a one-eyed racing driver managed to complete an entire season with only one incident[1], during seven laps, the driver failed to see and react to a penalty flag that was being waved on his normally sighted left side. To date, the literature contains no evidence of an increased risk of crashing among one-eyed drivers and that literature thus is unhelpful regarding the visual requirements of a racing driver[1]. Beyond the juridical and human rights issues, it is essential to determine if monocular drivers represent a potential risk to themselves and also to other racing drivers or even spectators at a race.

To our knowledge, no study has explored the impact of monocularity on driving in a racing condition. The only paper dealing with this issue is that of Westlake [1]. However, as mentioned by the author, monocularity has been studied only in the conventional driving research field. Indeed, the literature on performance and safety of monocular drivers has largely focused on studies of commercial drivers (eg truck, delivery vehicle, taxi, bus) and have reported conflicting results. For example, some research has found that monocular drivers have more crashes and convictions [2–4] and poorer driving performance [5]. Other studies have observed that, compared with control groups or a national reference group, drivers with real or simulated monocular visual field loss have equivalent accident or conviction rates and driving performance [6–8]. Furthermore, McKnight et al. [9] assessed 40 monocular and 40 binocular commercial drivers and found no differences with respect to visual search, lane placement, clearance judgment, gap judgment, hazard detection, or information recognition. However, these authors did find that monocular drivers were less adept at reading signs at a distance during both daytime and nighttime driving than were binocular drivers. Furthermore, the definition of "monocularity" has varied widely in the literature of conventional driving, ranging from drivers who have a total absence of function in one eye, to drivers who have visual function in one eye below the minimum level for having a license. Sometimes, no definition is provided. Thus, the impact of monocularity on conventional driving has not been exhaustively addressed. Finally, it is difficult to extrapolate the studies that have been performed on conventional driving to motorsport, which is a more demanding activity during which drivers are taking risks in a competitive context.

The lack of consensus on the impact of monocularity on driving is relatively surprising given the importance of visual inputs for this activity. Indeed, monocularity causes three important deficits that can affect driving. The first deficit concerns the reduction of the visual field. A monocular individual has a peripheral field deficit of 20 to 40 degrees temporally. Thus, monocular drivers must move their head or eye to obtain information on the temporal side of their non-functional eye. The impact on driving strongly depends on the importance of this information based on the context and the ability of the driver to adapt effectively their visual comportment to retrieve that information. The second deficit is the loss of binocular summation. It has been observed that there exists a superiority of binocular over monocular visual performance with fine central acuity being further enhanced through binocular summation [10]. Although it has been demonstrated in early enucleated patients and in subjects who are functionally monocular, that the contrast sensitivity [11,12] or visual acuity of the functional eye [13,14] is better than the visual functioning of the better eye of normal binocular subjects, the performance of monocular individuals remains inferior to those with binocular vision [14,15]. Finally, monocular drivers have a poorer perception of depth than binocular individuals [9],[14]. This may not be as important as the other deficits as the racetrack is a dynamic environment. Dynamic stereopsis is weakly correlated with static stereopsis and is reduced with increasing angular velocity [16,17]. The distances and operating speeds of stereopsis suggest that, in many circumstances, stereopsis would not be useful to the race driver. In addition, the driver can use a number of other indicators to assess the perception of depth,

such as the apparent size of the objects and their expansion over time, shadows, and the space between the front of the car they are driving and the car in front [1].

Although monocular drivers could be potentially seen as high-risk drivers, to date, no study has been conducted to determine if monocularity could be responsible for an objective decrease in visual performance during race driving and could impact safety. In addition, no study has evaluated the impact of a monocular generalized reduction in vision (MRV), such as that which occurs in patients with congenital amblyopia. The present study was performed to address the impact of monocularity and MRV on racing by simulating monocular visual field defects in a group of racing drivers to assess the impact of these defects on driving performance. Because it is difficult to perform these assessments in a standardized and safe way during real driving, we used a sophisticated driving simulator.

We conducted two experiments to evaluate the impact of monocularity and MRV on racing performance while driving. To our knowledge, this is the first study to do so. We predicted that under identical conditions, monocular drivers and drivers with monocularly reduced vision would be at higher risk for accidents and have longer reaction times than binocular drivers with normal visual function in both eyes.

## General method

### Overview

The study was conducted at the FFSA Autosport Academy (Le Mans, France) and at the Institut de la Vision, Paris, France. The study was approved by the Ethics Committee of the French Society of Ophthalmology and adhered to the tenets of the Declaration of Helsinki. All participants, or parents of minor participants, gave written informed consent to participate. No participants received compensation for participating. Seventy-five participants were recruited. All participants were professional racing drivers or students at the FFSA Autosport Academy. No exclusion criteria, such as demographic requirements, were applied when recruiting participants. The data from some participants were excluded from analysis because these participants did not complete all the tasks or because of failures in data recording.

### Visual assessment

A visual assessment was performed to select only visually non-impaired participants and to be sure that no visual pathologies could affect the results of the studies. The assessment included a complete medical and ocular history, tests of binocular and monocular visual acuity (ETDRS chart in Metrovision), colour perception (Ishihara plates), contrast sensitivity (Metrovision), visual field (kinetic perimetry using a Goldmann perimeter and a III/4e stimulus), refraction (TOPCON KR-800S), and spectacle lens and frame measurements (Essilor CLE 60). We also determined the sensory eye dominance to select the eye to be experimentally impaired (Red Lens test) and to determine which Ryser filter decreased the visual acuity of the dominant eye to 0.5 log MAR.

### Apparati

**Simulation of visual impairments.** Three visual conditions were employed in the study: baseline normal visual acuity (≤0 logMAR visual acuity) in both eyes, reduced vision (0.5 logMAR visual acuity) in the dominant eye, and monocular vision. The visual acuity restrictions were selected because they are most commonly cited as resulting in impaired driving performance [18]. MRV was achieved by placing a Ryser filter either on one lens of a pair of protective glasses (if the driver did not normally wear glasses) or on one lens of the driver's personal

glasses. Monocular vision was produced by placing an occlusive patch over the participant's dominant eye.

**Driving simulator.** Streetlab's fixed-base driving simulator (Paris, France) was used in this study. The driving simulation was handled by SCANeR™ studio Software and displayed on an immersive visual system consisting of three full HD LCD 65" screens that provided a forward field of view of 180 degrees. The simulator was equipped with a fixed-base car seat, a Fanatec force-feedback steering wheel, Fanatec braking and throttle pedals, and speakers. Both auditory feedback (i.e., engine noise) and tactile feedback (i.e., when contact was made with the curb in a corner the steering wheel jerked) were provided. Two video cameras were used to provide recordings of the driver's sessions. The driving parameters were recorded at the frequency of 60 Hz. The virtual driving environment was the former racetrack in Barcelona-Catalunya, Spain, and the vehicle being driven was a GP2 type car.

## Procedure

Participants attended two testing sessions. The first session was the selection session, during which potential participants completed the visual assessment and a Motion Sickness History Questionnaire MSHQ [19] to prevent unnecessary exposure to the driving simulation. The total duration of this session was 1h.

The second session was the driving simulator assessment. A 10–15 minutes practice drive preceded the test drive to allow the participant to become acquainted with the driving simulator and the track. The examiner issued a set of instructions to each subject before the run. Each racing driver was tested just once under each of the three visual conditions in a pseudo-random order. Participants could have a break whenever they wanted, so as to avoid Simulator Adaptation Syndrome (SAS). The total duration of the driving simulator session was around 1h.

## Statistical analysis

The minimum number of participants required was determined by an a priori power analysis. Our objective was to compare the occurrence of accidents by visual condition and report the estimated ORs using a logistic regression (the analysis was modified during the revision process). To determine the number of observations required per group (n), we assumed that the accident rate expected in the control condition (Baseline) would be 5% and 20% in the monocular condition with a power of at least 90% and an α of .05. We collected 100 observations per group, or a minimum number of 17 participants, taking into account the number of tests per driving situation.

A Generalized Linear Mixed Model (GLMM) with a logit link function was applied, using an R package named lme4 [20], in R [21] to study the likelihood (odds ratio) of collision risk given the visual condition of the racing drivers. The GLMM was used to account for the possible unobserved heterogeneity caused by repeated measures from the same individuals. The binary response variable was the involvement in an accident whereby 1 = accident and 0 = no accident. We set the normal vision condition as the reference condition. We assessed overdispersion for every model (the sum of the squared Pearson residuals should be $\chi 2$ distributed) [22].

We also analyzed driver reaction time in crash avoidance using two-way repeated measures ANOVA. Data were controlled for normality, homogeneity of variances, and sphericity to insure adequate tests. In cases where the assumption of sphericity was violated, the Greenhouse-Geisser correction was applied. A threshold of $p<.05$ was considered significant.

Dunnett's and Newman–Keuls post-hoc multiple comparison tests were conducted to identify where differences among means existed.

We analyzed the relationships between reaction times and accident rate for each visual condition, regardless of the type of situation, using Pearson's correlation.

## Experiment 1

The purpose of Experiment 1 was to investigate how simulated recent monocularity or MRV affects driving performance and particularly the detection of hazardous situations in front of the driver, within a horizontal field of vision of 120˚.

### Method

**Participants.**   31 participants initially were recruited for the experiment; however, three were withdrawn during testing due to symptoms of SAS. It also was necessary to exclude from analysis 10 participants with incomplete data (absence of data in at least 1 condition) or irrelevant data. The final sample was composed of 18 racing drivers ranging in age from 14 to 36 years (mean: 21.88 ± 7.47 years; 1 F, 17M). All participants were amateur or professional racing drivers with a mean of 6.8 ± 4.8 years of experience in AutoSport.

**Driving scenarios.**   Three different dynamic hazardous situations were developed according to the impaired-eye side (Fig 1). All three require the drivers to detect and react to a sudden intrusion of an opponent's racing car into the participant's trajectory in a turn. All opposing cars start from the edge of the track, within a visual field of 120 degrees maximum when the driver looks straight ahead. In the first situation, the turning direction (the direction where the drivers must look) and the location of the entering car are congruent with the visual impairment. We call this situation congruent-congruent (CC). In the second setting, the turning direction is congruent with the visual impairment, but the location of the entering car is incongruent with the visual impairment. We term this situation congruent-incongruent (CI). In the third situation, the turning direction (the direction where the drivers must look) is incongruent with the visual impairment, but the location of the entering car is congruent with the visual impairments. We call this situation incongruent-congruent (IC).

To prevent any anticipatory behavior, billboards hide the entering cars from the view of the drivers. These billboards have been placed at the inside and outside of every turn. Throughout the driving, the participant follows an opponent car with a fixed-time headway of 4s that cannot be overtaken. They also are followed by an opponent car with a fixed-time headway of 1.5s. This immersion process gives the impression to the driver of being in a race.

Six potential hazards were placed at different locations of the track for each of the three scenarios (CC, CI, IC), for 18 randomized situations. To experience all of the situations, the drivers must complete 15 laps. Six versions of the scenarios (three for each blind side) were generated to counterbalance the presentation order of the challenging events and so that each repeated condition had a different situation.

### Results

Table 1 lists the visual characteristics of the 18 tested subjects with complete data. No subject had any visual pathology or visual abnormality. All had normal visual acuity overall (mean: -0.23 ± 0.06 logMAR), and their MRV reached the desired visual acuity level (mean: 0.49 ± 0.04 logMAR).

**Driving performance.**   We evaluated 18 participants who passed all three visual conditions. In each condition, drivers were confronted with three types of dangerous situations. Six tests were proposed for each dangerous situation. Eliminating missing data related to the failed

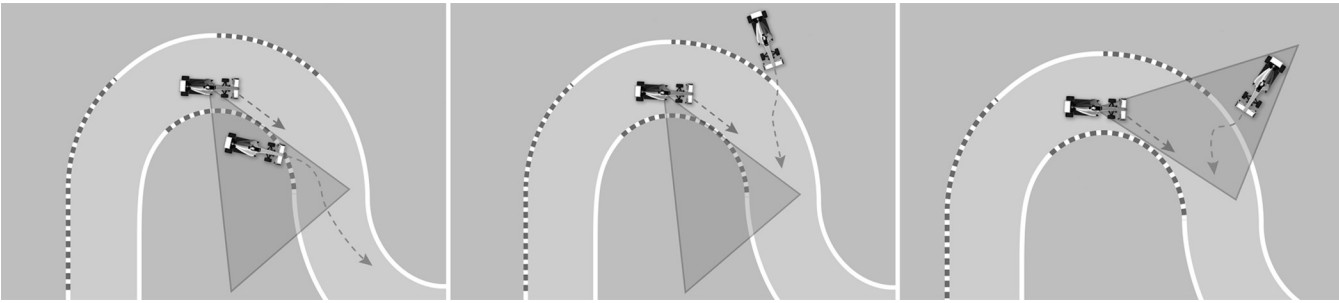

**Fig 1. The three hazardous situations, respectively CC, CI and IC situations.** The triangular shape corresponds to the blind area of the drivers.

or incorrectly performed tests, we collected 232 observations for the CC situations, 220 observations for the CI situations and 214 observations for the IC situations.

As a first step, we conducted a series of GLMMs to determine if multiple visual conditions (monocularity or MRV) were predictive of an accident and the odds of a collision happening given the visual condition. We conducted a GLMM for each potential hazard situation; i.e, CC, CI and IC (Table 2). The analysis indicated the absence of significant overdispersion for CC situations model ($\chi^2$ = 228.567; ratio = 1.002; rdf = 228; p = .477); for IC situations model ($\chi^2$ = 204.360; ratio = 0.973; rdf = 210; p = .597); and for CI situations model ($\chi^2$ = 203.934; ratio = 0.944; rdf = 216; p = .712).

For the CC situations, monocularity was the only significant condition in the model (p = .024). Model parameters showed that racing drivers in the monocular condition have 2.1 (95% CI 1.112–4.111) greater odds of having a collision than racing drivers in the Baseline binocular condition. For CI situations, the GLMM showed that neither monocularity nor MRV was

Table 1. Visual characteristics of tested subjects.

| Participants | Age (yr) | Visual Acuity (logMAr) | | | Visual Field Diameter Goldman III/4e | | Deficit within Central 10 Degrees | Contrast Sensitivity Abnormality | Color Vision Abnormality | Visual Acuity with Ryser |
|---|---|---|---|---|---|---|---|---|---|---|
| | | left | Right | Booth | Horizontal | Vertical | | | | |
| RD1 | 20 | -0,30 | -0,26 | -0,28 | 176 | 132 | no | no | no | 0,46 |
| RD2 | 28 | -0,28 | -0,22 | -0,26 | 174 | 119 | no | no | no | 0,50 |
| RD3 | 27 | -0,16 | -0,18 | -0,26 | 176 | 120 | no | no | no | 0,52 |
| RD4 | 31 | -0,20 | -0,26 | -0,20 | 176 | 123 | no | no | no | 0,52 |
| RD5 | 30 | 0,02 | -0,10 | -0,14 | 176 | 121 | no | no | no | 0,50 |
| RD6 | 36 | -0,28 | -0,28 | -0,30 | 176 | 112 | no | no | no | 0,56 |
| RD7 | 21 | -0,10 | -0,28 | -0,24 | 174 | 123 | no | no | no | 0,50 |
| RD8 | 17 | -0,26 | -0,24 | -0,28 | 173 | 120 | no | no | no | 0,50 |
| RD9 | 28 | -0,26 | -0,28 | -0,26 | 172 | 111 | no | no | no | 0,52 |
| RD10 | 34 | 0,12 | 0,06 | -0,06 | 174 | 120 | no | no | no | 0,50 |
| RD11 | 16 | -0,26 | -0,28 | -0,30 | 175 | 122 | no | no | no | 0,50 |
| RD12 | 15 | -0,16 | -0,20 | -0,24 | 176 | 124 | no | no | no | 0,46 |
| RD13 | 15 | -0,10 | -0,16 | -0,18 | 176 | 119 | no | no | no | 0,50 |
| RD14 | 15 | -0,24 | -0,26 | -0,30 | 171 | 122 | no | no | no | 0,40 |
| RD15 | 17 | -0,08 | -0,06 | -0,16 | 171 | 118 | no | no | no | 0,52 |
| RD16 | 16 | -0,26 | -0,22 | -0,30 | 172 | 118 | no | no | no | 0,54 |
| RD17 | 16 | -0,22 | -0,20 | -0,22 | 170 | 119 | no | no | no | 0,44 |
| RD18 | 15 | -0,2 | -0,2 | -0,2 | 171 | 110 | no | no | no | 0,44 |

**Table 2. Parameter estimates from the three GLMMs for the 3 driving situations related to the occurrence of collisions for each visual condition.**

| Parameter | Estimate | SE | Z value | p | OR | 95% CI low | 95% CI high |
|---|---|---|---|---|---|---|---|
| **Model for CC Situations** | | | | | | | |
| Intercept | -0,701 | 0,247 | -2,839 | 0,001 | | | |
| Baseline vision | 0,00 | 0,00 | | | | | |
| MRV | 0,484 | 0,335 | 1,447 | 0,148 | 1,623 | 0,845 | 3,153 |
| Monocular | 0,751 | 0,333 | 2,258 | 0,024 | 2,119 | 1,112 | 4,111 |
| **Model for CI Situations** | | | | | | | |
| Intercept | -0,710 | 0,282 | -2,514 | 0,012 | | | |
| Baseline vision | 0,00 | 0,00 | | | | | |
| MRV | 0,122 | 0,361 | 0,337 | 0,736 | 1,129 | 0,556 | 2,310 |
| Monocular | 0,286 | 0,352 | 0,812 | 0,417 | 1,331 | 0,668 | 2,679 |
| **Model for IC Situations** | | | | | | | |
| Intercept | -1,849 | 0,360 | -5,136 | 0,0001 | | | |
| Baseline vision | 0,00 | 0,00 | | | | | |
| MRV | 1,001 | 0,432 | 2,314 | 0,021 | 2,720 | 1,188 | 6,587 |
| Monocular | 1,129 | 0,428 | 2,637 | 0,008 | 3,094 | 1,368 | 7,456 |

SE: standard error; OR: odds ratio; CI: confidence interval.

significant. Finally, for IC situations, both monocularity and MRV were significant (p = .008 and p = .021, respectively). Model parameters showed that recently monocular racing drivers had 3.094 (95% CI 1.368–7.456) greater odds of having a collision compared with the baseline condition, and racing drivers with MRV had 2.72 (95% CI 1.188, 6.587) times greater odds of having a collision compared with the baseline visual condition.

**Reaction time.** Mean reaction times for the three hazard situations are presented in the Fig 2. The two-way repeated measures ANOVA showed significant main effects of the visual condition, $F(2, 34) = 5.587$, p = .008, and a tendential effect of the hazard situation, $F(2, 34) = 2.695$, p = .082. The interaction effect was non-significant, $F(4, 68) = .9559$, p = .437.

We performed separate one-way repeated ANOVAs for each hazard situation and then conducted multiple comparisons to assess the simple main effect of each visual condition. For the CC hazard situation, the ANOVA revealed a significant main effect of visual condition, $F(2, 34) = 3.630$, p = .049. Dunnett's multiple comparison showed a significantly longer RT under the monocular visual condition (800 ms) than under the baseline visual condition (736 ms, p = .029). We did not observe a difference in RT between the MRV condition and the baseline visual condition (p = .837). Furthermore Newman-Keuls post-hoc test showed a significant longer RT under the monocular visual condition (800 ms) compared with the MRV condition (748 ms, p = .049). For the CI hazard situation, the effect of visual condition was not significant, $F(2,34) = 1.110$ p = .341. For the IC hazard situation, the analysis revealed a significant main effect of both visual conditions, $F(2, 34) = 4,407$, p = .019, with Dunnett's multiple comparisons indicating that RT was significantly longer under the monocular visual condition (874 ms) than baseline visual condition (748 ms, p = .015) and tended to be significantly longer under the MRV condition (846 ms) than baseline visual condition (748 ms, p = .064). Newman-Keuls post-hoc test failed to reveal a statistical difference between the monocular visual condition and the MRV condition (p = .531).

**Correlations between reaction times and accident rate.** As the variables were normally distributed, we used a Pearson correlation

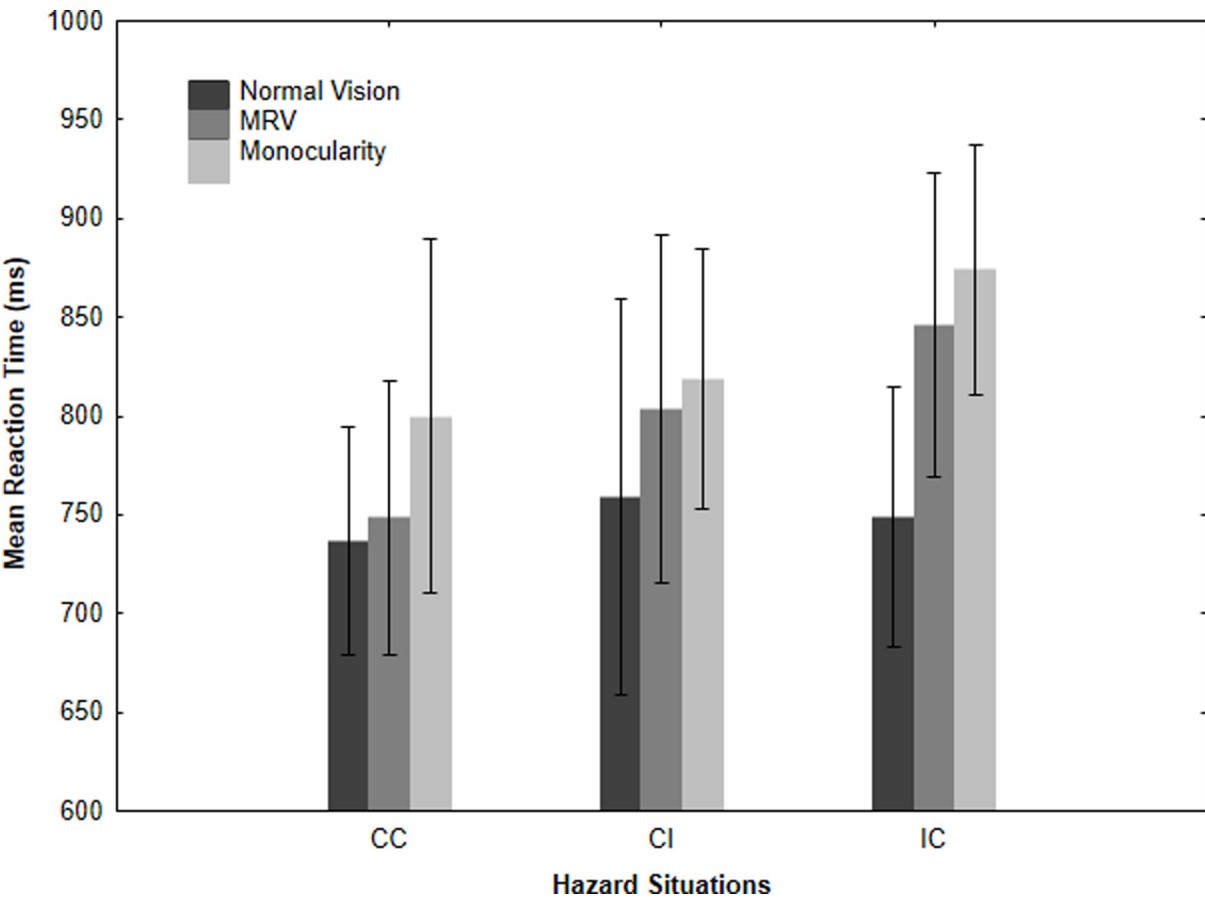

**Fig 2. The effects of visual conditions on mean reaction times for each hazard situation.** The error bars represent the 95% confidence interval.

Regarding the baseline visual condition, the results of the Pearson correlation indicated that there was no significant association between reaction times and accident rates, (r = .28, p = .26). Regarding the MRV visual condition, the results of the Pearson correlation indicated that there was a significant positive association between reaction times and accident rates, (r = .64, p = .004). Finally, regarding the monocular visual condition, the results of the Pearson correlation indicated that there was a significant positive association between reaction times and accident rates, (r = .48, p = .044).

## Experiment 2

The purpose of Experiment 2 was to investigate how simulated recent monocularity or MRV affects driving performance and particularly the detection of hazard situations entering the horizontal field of vision from the rear.

### Method

**Participants.** 44 participants were recruited for the second experiment. Four were discarded due to early symptoms typical of SAS. It also was necessary to exclude from the analysis nine participants with incomplete data. The final sample thus was composed of 31 racing drivers, all men, ranging in age from 15 to 48 years (mean: 25.16 ± 9.91 years). All participants

were amateur or professional racing drivers with a mean of 8.46 ± 7.82 years of experience in Autosport. 15 were formula racing drivers, six were kart-racing drivers, six were GT racing drivers, one was a rally-racing driver and three were driving instructors.

**Driving scenarios.** Two different dynamic hazardous situations were developed (Fig 3), both of which require the drivers to detect and react to the sudden intrusion of an opponent's racing car into the participant's trajectory when entering a turn. In those situations, the driver is overtaken by an opponent vehicle as the driver's car approaches a turn. The opponent vehicle appears suddenly, accelerates to overtake the driver, and enters the visual field of the driver at one of two visual angles: 75˚ or 60˚, assuming that the driver is looking straight ahead in the direction their vehicle is moving. When it reaches one of the above-mentioned angles, the opponent vehicle stabilizes its speed to stay at this angle position until the driver breaks and turns the steering wheel. To prevent the presence of the opponent car in the visual field from being detected by the driver using their near visual field (less than 60˚), the rear-view mirrors have been obstructed visually. This provision is consistent with the report by drivers engaged with an opponent that they sometimes neglect to look in their rearview mirror and can be surprised by the intrusion of another opponent.

In the first situation, the turning direction (the direction where the driver must look) and the location of the entering car are congruent with the visual impairment. We call this situation congruent-congruent (CC). In the second situation, both the turning direction of the driver's car and the location of the entering car are incongruent with the visual impairment. We call this situation incongruent-incongruent (II). Different locations were chosen for both CC and II situations according to the side of the impaired eye.

The session consisted of eight trials for the CC and four trials for the II randomized situations located at different turns of the circuit. Six versions of each situation (three for each blind side) were generated to counterbalance the presentation order of the challenging events and so that each repeated condition had a different situation.

## Results

Table 3 lists the visual characteristics of the 31 participants. No participant had any visual pathology or visual abnormality (mean visual acuity: -0.23 ± 0.05 logMAR) and their visual acuity for MRV reached the desired visual acuity level (mean: 0.49 ± 0.04 logMAR).

**Driving performance.** We evaluated 31 participants who passed all three visual conditions. In each condition the drivers were confronted with two types of dangerous situations. Eight tests were proposed for the CC situations and four tests for the II situations. If we eliminate the failed or incorrectly performed tests, we collected 658 observations for the CC condition and 325 observations for situation II.

We conducted a series of GLMMs to determine if monocularity or MRV were predictive of an accident and the odds of a collision occurring, given the specific visual condition. A GLMM was conducted for both the CC and II hazard situations (Table 4). The analysis indicate the absence of significant overdispersion for CC situations model ($\chi^2$ = 590.933; ratio = 0.903; rdf = 654; p = .963) and for II situations model ($\chi^2$ = 242.486; ratio = 0.973; rdf = 320; p = .999).

Monocularity in the CC situations model was the only significant condition (p<.0001). Model parameters showed that racing drivers in the monocular condition have 6.493 (95% CI 3.911–11,133) greater odds of having a collision compared with racing drivers in the baseline condition.

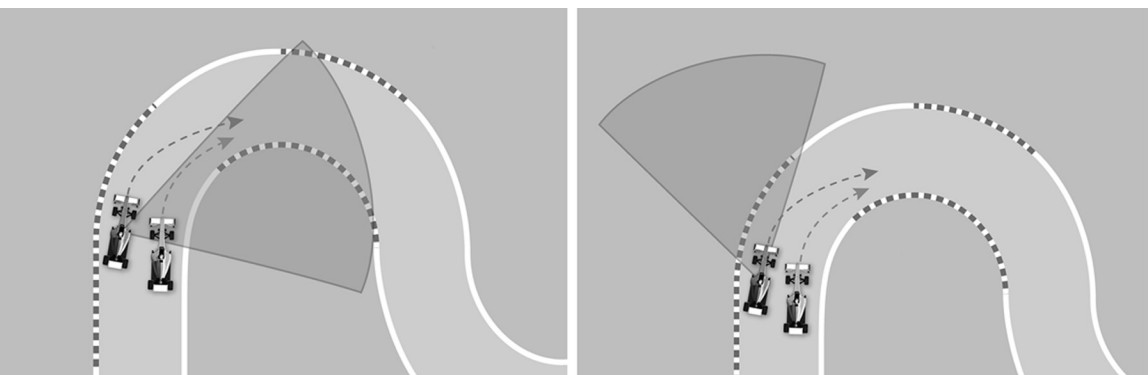

**Fig 3. The two hazardous situations, respectively CC and II situations for experiment 2.** The triangular shape corresponds to the blind area of the driver.

## Discussion

This study investigated the impact of simulated recent monocularity and MRV on simulated racing driving performance. The results of the study demonstrate that driving under racing conditions with simulated recent monocularity affects driving performance. Specifically, the results of the two experiments show that, compared with the baseline condition, drivers under the recent monocular condition are 2.1 to 6.5 times more likely to collide with target vehicles, depending on the driving situation. The driving situations that pose problems in monocular conditions are those in which the target to be detected and avoided appears on the blind-eye side (CC and IC situations for experiment 1 and CC for experiment 2). The maximum risk in all situations is reached for the situation CC in experiment 2. In this situation, the target initially is outside the visual field and rapidly enters the field of view. The results therefore seem to indicate that the more the target to be detected is in the peripheral vision on the blind side, the greater is the risk of collision. This is explained by the fact that the more the target to be detected is in the periphery on the blind side, the later it will be detected, leaving less time and, therefore, less chance for the driver to avoid a collision. The increase in reaction time, observed for drivers in the monocular vision condition (the IC [126 ms] and CC [64 ms] situations in experiment 1), seems to confirm the above results. In addition, correlation analysis allows us to observe that the longer the reaction time increases in this visual condition, the greater the risk of an accident. On the other hand, the results of the CI situation of experiment 1, and the II situation of experiment 2, characterized by the fact that the targets appear on the "healthy" side of the eye, show that these situations do not pose a problem for drivers under the monocular condition.

The simulated recent MRV condition seems to have a limited impact on driving. In the IC situation, there is both an increased risk of collision, where drivers are 2.7 times more likely to be involved in a crash, and a reaction time which tended to be longer than under normal vision conditions (98ms). It is possible that the drivers in this setting optimize their trajectory by moving their head towards the tangent point or the apex. By this head movement, it is possible that the appearance of the vehicle is outside their visual field and so would consequently be detected later, making the collision more difficult to avoid. Nevertheless, correlation analysis allows us to observe that, for the monocular visual condition, the longer the reaction time increases, the greater the risk of an accident, whereas this is not the case in baseline visual condition.

**Table 3. Visual characteristics of tested subjects.**

| Participants | Age (yr) | Visual Acuity (logMAr) | | | Visual Field Diameter Goldman III/4e | | Deficit within Central 10 Degrees | Contrast Sensitivity Abnormality | Color Vision Abnormality | Visual Acuity with Ryser |
|---|---|---|---|---|---|---|---|---|---|---|
| | | left | Right | Booth | Horizontal | Vertical | | | | |
| RD1 | 17 | -0,28 | -0,22 | -0,30 | 174 | 123 | no | no | no | 0,56 |
| RD2 | 29 | -0,26 | -0,20 | -0,26 | 176 | 107 | no | no | no | 0,52 |
| RD3 | 21 | -0,08 | -0,16 | -0,20 | 171 | 98 | no | no | no | 0,46 |
| RD4 | 46 | -0,18 | -0,20 | -0,26 | 176 | 121 | no | no | no | 0,52 |
| RD5 | 15 | -0,20 | -0,26 | -0,26 | 176 | 124 | no | no | no | 0,56 |
| RD6 | 16 | -0,26 | -0,26 | -0,30 | 176 | 124 | no | no | no | 0,54 |
| RD7 | 25 | -0,28 | -0,28 | -0,28 | 173 | 118 | no | no | no | 0,50 |
| RD8 | 32 | 0,020 | 0,02 | -0,08 | 175 | 121 | no | no | no | 0,56 |
| RD9 | 23 | -0,02 | -0,12 | -0,22 | 168 | 117 | no | no | no | 0,52 |
| RD10 | 21 | -0,26 | -0,22 | -0,28 | 176 | 123 | no | no | no | 0,50 |
| RD11 | 15 | -0,10 | -0,16 | -0,18 | 176 | 119 | no | no | no | 0,50 |
| RD12 | 35 | -0,24 | -0,28 | -0,30 | 174 | 117 | no | no | no | 0,50 |
| RD13 | 19 | -0,28 | -0,28 | -0,28 | 171 | 121 | no | no | no | 0,50 |
| RD14 | 17 | -0,10 | -0,16 | -0,20 | 173 | 115 | no | no | no | 0,52 |
| RD15 | 19 | -0,14 | -0,16 | -0,20 | 171 | 117 | no | no | no | 0,50 |
| RD16 | 36 | -0,24 | -0,24 | -0,24 | 175 | 124 | no | no | no | 0,50 |
| RD17 | 21 | -0,18 | -0,18 | -0,24 | 171 | 117 | no | no | no | 0,54 |
| RD18 | 26 | -0,06 | -0,12 | -0,22 | 175 | 121 | no | no | no | 0,62 |
| RD19 | 35 | -0,14 | -0,02 | -0,14 | 171 | 122 | no | no | no | 0,50 |
| RD20 | 16 | -0,22 | -0,20 | -0,22 | 170 | 119 | no | no | no | 0,44 |
| RD21 | 31 | 0,06 | -0,20 | -0,28 | 166 | 124 | no | no | no | 0,56 |
| RD22 | 23 | -0,28 | -0,22 | -0,30 | 170 | 117 | no | no | no | 0,54 |
| RD23 | 15 | 0,02 | -0,10 | -0,10 | 172 | 123 | no | no | no | 0,48 |
| RD24 | 15 | -0,26 | -0,20 | -0,26 | 170 | 117 | no | no | no | 0,60 |
| RD25 | 20 | -0,18 | -0,14 | -0,22 | 176 | 123 | no | no | no | 0,48 |
| RD26 | 22 | -0,20 | -0,06 | -0,28 | 173 | 112 | no | no | no | 0,46 |
| RD27 | 48 | -0,20 | 0,06 | -0,20 | 168 | 105 | no | no | no | 0,40 |
| RD28 | 47 | -0,16 | -0,16 | -0,22 | 170 | 90 | no | no | no | 0,50 |
| RD29 | 24 | -0,24 | -0,26 | -0,28 | 170 | 115 | no | no | no | 0,54 |
| RD30 | 15 | -0,16 | -0,20 | -0,22 | 170 | 114 | no | no | no | 0,52 |
| RD31 | 36 | -0,20 | +0,12 | -0,26 | 173 | 115 | no | no | no | 0,50 |

The results of this study, and specifically in experiment 2, could be explained by the fact that reduced central acuity (in one eye) is not as important in racing as loss of peripheral vision and that information obtained from the peripheral field is useful or at least relevant for race car driving. Peripheral vision has characteristics that are very different from central vision. Firstly, peripheral vision makes it possible to assess the entire visual scene, especially its spatial layout, more quickly than with eye movements (and, therefore, even more quickly than with head movements). The basic understanding of a scene can be completed in less than 100ms [23,24], which would be impossible without peripheral vision [25]. It also has the capacity to extract properties for a wide variety of stimuli, such as the size of the object [26], object orientation [27], and even the average emotion of a group of faces [28,29]. It also is possible to extract the mean pedestrian heading [30] or detect movement [31] through peripheral vision that can be very useful in the context of driving. The evaluation of the average characteristic

**Table 4. Parameter estimates from the two GLMMs for the 2 driving situations related to the occurrence of collisions for each visual condition.**

| Parameter | Estimate | SE | Z value | p | OR | 95% CI low | 95% CI high |
|---|---|---|---|---|---|---|---|
| **Model for CC Situations** | | | | | | | |
| Intercept | -2,041 | 0,274 | -7,448 | 0,0001 | | | |
| Baseline vision | 0,00 | 0,00 | | | | | |
| MRV | 0,376 | 0,273 | 1,377 | 0,168 | 1,457 | 0,851 | 2,523 |
| Monocular | 1,87 | 0,263 | 7,102 | 0,0001 | 6,493 | 3,911 | 11,133 |
| **Model for II Situations** | | | | | | | |
| Intercept | -1,554 | 0,348 | -4,463 | 0,0001 | | | |
| Baseline vision | 0,00 | 0,00 | | | | | |
| MRV | 0,083 | 0,351 | 0,239 | 0,811 | 1,087 | 0,538 | 2,211 |
| Monocular | 0,112 | 0,360 | 0,312 | 0,755 | 1,119 | 0,542 | 2,321 |

value of a group of similar objects (pedestrians, bicycles, etc.), commonly called ensemble perception, is accomplished at the expense of the ability to discriminate more finely the information specific to each object taken individually [25].

Thus, peripheral vision allows the driver to detect elements entering the visual field, acquire information on these elements and, if needed, direct the eyes to process this information more finely. In addition, it has been shown that the identification of objects on the periphery is largely facilitated via pre-saccadic attention [32]. Thus, the analysis of the environment via the peripheral vision does not require perfect central vision. A car entering in the visual field of the driver will be very quickly detected and the driver will certainly not even have to look towards the object to infer that it is a car, a truck or a motorcycle. This could explain why the simulated recent MRV is less affected than the recent monocularity condition. Our results, especially those of experiment 2, suggest that the preserved peripheral visual field in drivers with recent MRV are able to perceive the appearance of a car entering their visual field and, thus, avoid a collision. On the other hand, the absence of peripheral vision in a simulated recent monocular condition strongly penalizes the driver.

The results of this study show that acute monocularity can impact the safety of racing drivers. Even though the participants in this study were professional drivers in racing conditions, we believe that it is reasonable to extrapolate these results to conventional driving. Indeed, the configurations of the situations tested in our experiments may well apply to an on-road driving condition. For example, they may represent an overtaking situation or a sudden insertion situation into the vehicle's trajectory at a crossroads or the untimely crossing of a pedestrian outside a normal crosswalk. As indicated in the introduction that there is a lack of consensus in the literature on the impact of monocularity on safety and driving performance. This is partly due to the operational definition of monocularity that varies among studies ranging from no function at all in one eye to one eye with reduced visual function to some extent (usually visual acuity). However, we see the importance of defining the phenomenon precisely in this study as our results show that the impact is not at all the same between a recent monocular condition and a recent MRV condition.

Nevertheless, we must remain cautious in this generalization as the dynamics between sports and classic driving are not the same. Thus, it would be interesting to reproduce the design of our study in this context of open-road driving.

There are certain limitations to our study. Firstly, as the study drivers did not have a pre-existing visual impairment, the results relate only to individuals who either have suddenly become monocular or have experienced a sudden monocular generalized visual loss. Such individuals have not had time to adapt to the vision alteration by putting in place behavioral

strategies to deal with the visual impairment. It would be interesting to evaluate drivers with long-standing or even congenital monocularity or MRV and who have had time to adapt to their visual disability to see if they are using strategies to deal effectively with these situations.

Another limitation of our study relates to the methodology used in experiment 2. To meet the objectives of this experiment, we forced the drivers to drive without their mirrors. Indeed, we wanted to evaluate the impact of monocularity without the possibility of behavioral adaptation. This methodology has the virtue of placing the driver in a worst-case scenario but may be perceived as a little less environmentally realistic. However, it is quite possible, for various reasons, that a mirror may no longer be accessible to the driver (eg, a mirror that is broken, fogged, or useless due to the glare of the sun). It also is possible that, in some cases, the driver may not see an element in its wing mirror. Indeed, motor racing can involve an intense struggle between drivers. They are highly engaged and allocate a very important part of their attention to interactions with the vehicles in front of them, which they seek to overtake. This can lead to a phenomenon of inattentive blindness because their attention is strongly focused on another spot, event or object. The expression "inattentional blindness" was created by Arien, Mack and Irvin Rock [33] and popularized by the invisible gorilla test, conducted by Simons and Chabris [34]. In the case of inattention blindness, drivers can miss, for example, the overtaking maneuver of another car coming from behind them.

To improve the experience, it would be interesting to use a more ecological methodology to test each driver in a single condition with a single hazardous event. Indeed, in our study, we tested each driver in the three visual conditions and with 12 or 18 hazardous situations each time. However, these events are relatively rare while racing. Thus, the appearance of events would be less predictable. Nevertheless, the implementation of this method requires the involvement of a considerable number of drivers who are very hard-to-reach people. It could also be very interesting to perform a longitudinal study with drivers who have lost vision in one eye to see if they can compensate for their deficit and evaluate their adaptation time to recover performances as good as those of binocular drivers.

The present findings suggest that a sudden monocular reduction in vision with preservation of the peripheral field does not impact either race car driving performance or safety loss but that sudden monocularity has a great impact on driving performance and safety during car racing. Accordingly, we believe that further research should be conducted on monocular drivers who have had time to adapt to their disability to determine if adaptation is possible and sufficient to lead to an improved or even normal driving performance. In addition, future research should also look at how sudden and long-standing monocularity impacts driving for every day, non-professional drivers.

## Supporting information

**S1 Dataset. This is the dataset of experiment 1 for CC condition.**
(XLSX)

**S2 Dataset. This is the dataset of experiment 1 for CI condition.**
(XLSX)

**S3 Dataset. This is the dataset of experiment 1 for IC condition.**
(XLSX)

**S4 Dataset. This is the dataset of experiment 2 for CC condition.**
(XLSX)

**S5 Dataset. This is the dataset of experiment 2 for II condition.**
(XLSX)

**S6 Dataset. This is the dataset of experiment 1 for correlation analysis.**
(XLSX)

**S7 Dataset. This is dataset of experiment 1 for reaction time analysis.**
(XLSX)

## Acknowledgments

The author would like to thank Christopher Reeves for his helpful contribution in this Paper. The authors gratefully acknowledge all the member of FFSA Academy and more specifically Christophe Lollier for their technical assistance. The authors would also like to thanks jean-Jacques Issermann for his helpful assistance.

## Author Contributions

**Conceptualization:** Julien Adrian, José-Alain Sahel, Gérard Saillant, Bahram Bodaghi.

**Formal analysis:** Julien Adrian.

**Investigation:** Julien Adrian, Johan Le Brun.

**Methodology:** Julien Adrian, Johan Le Brun.

**Software:** Johan Le Brun.

**Supervision:** Julien Adrian, José-Alain Sahel, Gérard Saillant.

**Validation:** José-Alain Sahel, Gérard Saillant, Bahram Bodaghi.

**Writing – original draft:** Julien Adrian.

**Writing – review & editing:** Johan Le Brun, Neil R. Miller, José-Alain Sahel, Gérard Saillant, Bahram Bodaghi.

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
