## [Decision Letter · Decision Letter 0]

23 Aug 2019

PONE-D-19-19649

Implications of monocular vision for racing drivers

PLOS ONE

Dear Dr. Adrian,

Thank you for submitting your manuscript to PLOS ONE. After careful consideration, we feel that it has merit but does not fully meet PLOS ONE’s publication criteria as it currently stands. Therefore, we invite you to submit a revised version of the manuscript that addresses the points raised during the review process.

The manuscript has been reviewed by two experts in transportation safety and medical biostatistics. Based on the review results, some major issues need to be fully addressed in the revised manuscript. Particularly, some of the major issues are: (1) statistical analysis method issue as brought up in the comments; and (2) sample size issue of statistical analysis. Please completely address the aforementioned major issues along with other detailed comments from both reviewers in a thoroughly revised manuscript in order for the manuscript to be recommended for acceptance.

We would appreciate receiving your revised manuscript by Oct 07 2019 11:59PM. To enhance the reproducibility of your results, we recommend that if applicable you deposit your laboratory protocols in protocols.io, where a protocol can be assigned its own identifier (DOI) such that it can be cited independently in the future. For instructions see: http://journals.plos.org/plosone/s/submission-guidelines#loc-laboratory-protocols

We look forward to receiving your revised manuscript.

Kind regards,

Zhixia Li, Ph.D.

Academic Editor

PLOS ONE

Journal Requirements:

"I have read the journal's policy and some authors of this manuscript have the following

competing interests:

José-Alain Sahel:

Funding Sources: LabEx LIFESENSES (ANR-10-LABX-65), ERC Synergy

"HELMHOLTZ" (ERC Grant Agreement #610110), Banque publique d'Investissement

(Sightagain BPI-2014-PRSP-15), University-Hospital Institute “FOReSIGHT ((ANR-18-

IAHU-01)”, Foundation Fighting Blindness (C-CL-0912-0600-INSERM01; C-GE-0912-

0601-INSERM02).

Consultant: Pixium Vision; GenSight Biologics; SparingVision.

Personal Financial Interests: GenSight Biologics, Prophesee, Chronolife, Pixium

Vision, Tilak Healthcare, Sparing Vision.

Bahram Bodaghi is consultant for the Medical Commission of the FIA.

Neil R. Miller is study director for the QRK207 Clinical Treatment Trial for acute NAION

(Quark Pharmaceuticals) and consultant to the Regenera pharmaceutical company.

Gérard Saillant is President of the Medical Commission of the FIA.

The other authors do not have any conflicts of interest to disclose".

Additional Editor Comments (if provided):

The manuscript has been reviewed by two experts in transportation safety and medical biostatistics. Based on the review results, some major issues need to be fully addressed in the revised manuscript. Particularly, some of the major issues are: (1) statistical analysis method issue as brought up in the comments; and (2) sample size issue of statistical analysis. Please completely address the aforementioned major issues along with other detailed comments from both reviewers in a thoroughly revised manuscript in order for the manuscript to be recommended for acceptance.

Reviewers' comments:

Reviewer's Responses to Questions

**Comments to the Author**

1. Is the manuscript technically sound, and do the data support the conclusions?

Reviewer #1: Yes

Reviewer #2: Yes

2. Has the statistical analysis been performed appropriately and rigorously? 

Reviewer #1: Yes

Reviewer #2: No

3. Have the authors made all data underlying the findings in their manuscript fully available?

Reviewer #1: Yes

Reviewer #2: No

4. Is the manuscript presented in an intelligible fashion and written in standard English?

Reviewer #1: Yes

Reviewer #2: Yes

5. Review Comments to the Author

Reviewer #1: Interesting paper. I only have minor comments as the follows:

* All participants are non visually impaired. It is possible that the test drivers perform worse under monocular vision case because they are not used to rely on only one eye. The drivers that adapt to the monocular vision may perform much better. The authors discussed this issue in the conclusion section. I was wondering if the authors can give some recommendations on how we can improve the experiments and analyses in the future studies.

* In line 176, why was each driver tested only once in each condition? Multiple tests in each test might generate richer samples for eliminating test errors.

* In line 188, if every situation is a separate sample, how can we perform a statistically meaningful analysis?

* In line 231, how can a human driver keep a fixed headway from the preceding vehicle?

Reviewer #2: The authors designed two experiments to investigate whether monocular vision or a monocular generalized reduction in vision (MRV) impacts driving performance during racing. I do have some comments on the statistical analysis and wish the authors could address them:

1. In the first experiment, there were 18 participants. But how many observations were obtained from each participant? Why the observations from the same participant could be viewed as independent? A mixed logistic model and the ANOVA for repeated measurements may be more appropriate.

2. Also in the first experiment, how many participants were in CC, CI, and IC, respectively? On average, there were six subjects in each scenario. The authors need to justify why the sample size is sufficient to detect meaningful difference. A similar justification may also be needed for the second experiment

3. I wonder whether the authors could explain why the reaction time was not measured and compared in the second experiment?

4. In the discussion, the authors attempted to explain how monocular vision or MRV impacts driving performance during racing through reaction time. Is it possible that the authors could do some analysis to show the connection between reaction time and collision?

6. PLOS authors have the option to publish the peer review history of their article (what does this mean?). If published, this will include your full peer review and any attached files.

Reviewer #1: No

Reviewer #2: No

---

## [Author Response · Author response to Decision Letter 0]

27 Sep 2019

Response to reviewers

September 2019

In red are the response of the authors

As asked by the Editor, we include our updated Competing Interests statement in the response to reviewers’ letter. 

I have read the journal's policy and some authors of this manuscript have the following

competing interests:

José-Alain Sahel:

Funding Sources: LabEx LIFESENSES (ANR-10-LABX-65), ERC Synergy

"HELMHOLTZ" (ERC Grant Agreement #610110), Banque publique d'Investissement

(Sightagain BPI-2014-PRSP-15), University-Hospital Institute “FOReSIGHT ((ANR-18-

IAHU-01)”, Foundation Fighting Blindness (C-CL-0912-0600-INSERM01; C-GE-0912-

0601-INSERM02).

Consultant: Pixium Vision; GenSight Biologics; SparingVision.

Personal Financial Interests: GenSight Biologics, Prophesee, Chronolife, Pixium

Vision, Tilak Healthcare, Sparing Vision.

Bahram Bodaghi is consultant for the Medical Commission of the FIA.

Neil R. Miller is study director for the QRK207 Clinical Treatment Trial for acute NAION

(Quark Pharmaceuticals) and consultant to the Regenera pharmaceutical company.

Gérard Saillant is President of the Medical Commission of the FIA.

The other authors do not have any conflicts of interest to disclose

This does not alter our adherence to PLOS ONE policies on sharing data and materials.

Supporting Information has been added at the end of the manuscript.

S1 Dataset. This is the dataset of experiment 1 for CC condition.

S2 Dataset. This is the dataset of experiment 1 for CI condition.

S3 Dataset. This is the dataset of experiment 1 for IC condition.

S4 Dataset. This is the dataset of experiment 2 for CC condition.

S5 Dataset. This is the dataset of experiment 2 for II condition.

S6 Dataset. This is the dataset of experiment 1 for correlation analysis.

S7 Dataset. This is dataset of experiment 1 for reaction time analysis.

Responses to Reviewer #1:

* All participants are non visually impaired. It is possible that the test drivers perform worse under monocular vision case because they are not used to relying on only one eye. The drivers that adapt to the monocular vision may perform much better. The authors discussed this issue in the conclusion section. I was wondering if the authors can give some recommendations on how we can improve the experiments and analyses in the future studies.

Answer: 

We appreciate the reviewer’s question and have added in the discussion of the manuscript:

- Page 25-26 line 519-528:“To improve the experience, it would be interesting to use a more ecological methodology to test each driver in a single condition with a single hazardous event. Indeed, in our study we tested each driver in the three visual conditions and with 12 or 18 hazardous situations each time. However, these events are relatively rare while racing. Thus, the appearance of events would be less predictable. Nevertheless, the implementation of this method requires the involvement of a considerable number of drivers who are very hard-to-reach people. It could also be very interesting to perform a longitudinal study with drivers who have lost vision in one eye to see if they can compensate for their deficit and evaluate their adaptation time to recover performances as good as those of binocular drivers.”

We also made a correction page 18 line 375-376. “The session consisted of eight trials for the CC and four for the II randomized situations located at different turns of the circuit”.

* In line 176, why was each driver tested only once in each condition? Multiple tests in each test might generate richer samples for eliminating test errors.

We appreciate the reviewer’s question. We were only able to test each driver once in each visual condition because the vast majority of them are professional drivers who have a very busy schedule and therefore had only limited time to do the driving simulator test. It was therefore not possible to have more time with them or to review them after the test. Furthermore, testing multiple times in each condition might induce a learning effect, like strategies to avoid situations, that might bias the results. 

* In line 188, if every situation is a separate sample, how can we perform a statistically meaningful analysis?

We appreciate the reviewer’s question. The principle is to use each tested situation independently which takes a binary form of 0 or 1 depending on whether or not there has been an accident. To evaluate these data, we use statistics that model binary variables and are analytical models commonly used in epidemiology. For example. we had initially used a logistic regression but we eventually performed a GLMM at the request of reviewer 2.

* In line 231, how can a human driver keep a fixed headway from the preceding vehicle?

We appreciate the reviewer’s question. In our simulation experiments, the driver does not maintain the time headway. It is the front car that is driven by the simulator software that provides the 4s interval. No matter what the driver does, the car in front of him adapts by accelerating or braking to obtain a constant time headway of 4 seconds.

Responses to Reviewer #2:

Reviewer #2: The authors designed two experiments to investigate whether monocular vision or a monocular generalized reduction in vision (MRV) impacts driving performance during racing. I do have some comments on the statistical analysis and wish the authors could address them:

1. In the first experiment, there were 18 participants. But how many observations were obtained from each participant? Why the observations from the same participant could be viewed as independent? A mixed logistic model and the ANOVA for repeated measurements may be more appropriate.

We appreciate the reviewer’s question and comment. In experiment 1, we tested the subject for each visual condition 18 times, with 6 tests per type of situation (CC, CI and IC). In experiment 2, we tested the subject 12 times for each visual condition with 8 tests for the CC condition and 4 tests for condition II. A number of situations failed for various reasons, such as the loss of control of the car at the time of the event or the subject's poor positioning in the curve, which reduced the number of final situations. 

We initially considered the observations as independent for several reasons:

- Drivers have a very stereotypical behaviour unlike conventional driving. They are all trying to go as fast as possible by following the ideal trajectory on the circuit. 

- Each test situation has been implemented in a different turn and no two turns have the same characteristics (speed, angle, dynamics, grip...). 

- A driver is not homogeneous in his driving performance. Its performance is not the same according to the turn or sectors of the track. 

Having said this, we have followed the recommendations of the reviewer by adding a GLMM that corresponds to a logistic regression in which we control the effects related to the participants' test-retest. We used a Generalised Linear Mixed Model (GLMM) with a logit link function to account for the possible unobserved heterogeneity caused by repeated measures from the same individuals. The results of the GLMM have been inserted in the manuscript. 

On the other hand, for reaction times we had initially used an analysis of variance with repeated measurements.

Concerning the modification of the statistical analysis we added:

- Page 2 line 32. “Generalized Linear Mixed Models (GLMMs)”

- Page 2 line 36=5-37=6. “2.1 (95% CI 1.11 - 4.11, p=.024) to 6.5 (95% CI 3.91 – 11.13; p=.0001)”

- Page 9 line 191-204. “A Generalized Linear Mixed Model (GLMM) with a logit link function was applied, using an R package named lme4 [20], in R [21] to study the likelihood (odds ratio) of collision risk given the visual condition of the racing drivers. The GLMM was used to account for the possible unobserved heterogeneity caused by repeated measures from the same individuals. The binary response variable was the involvement in an accident whereby 1= accident and 0= no accident. We set the normal vision condition as the reference condition. We assessed overdispersion for every model (the sum of the squared Pearson residuals should be χ2 distributed)[22].”

- Page 13 line 272-281.” As a first step, we conducted a series of GLMMs to determine if multiple visual conditions (monocularity or MRV) were predictive of an accident and the odds of a collision happening given the visual condition. We conducted a GLMM for each potential hazard situation; i.e, CC, CI and IC (Table 2). The analysis indicated the absence of significant overdispersion for CC situations model (�2= 228.567; ratio= 1.002; rdf= 228; p= .477); for IC situations model (�2= 204.360; ratio= 0.973; rdf= 210; p= .597); and for CI situations model (�2= 203.934; ratio= 0.944; rdf= 216; p= .712).

- Page 13-14 line 283-286.”

Table 2. Parameter estimates from the three GLMMs for the 3 driving situations related to the occurrence of collisions for each visual condition.

Parameter Estimate SE Z value p OR 95% CI low 95% CI high

Model for CC Situations

Intercept -0,701 0,247 -2,839 0,001 

Baseline vision 0,00 0,00 

MRV 0,484 0,335 1,447 0,148 1,623 0,845 3,153

Monocular 0,751 0,333 2,258 0,024 2,119 1,112 4,111

Model for CI Situations 

Intercept -0,710 0,282 -2,514 0,012 

Baseline vision 0,00 0,00 

MRV 0,122 0,361 0,337 0,736 1,129 0,556 2,310

Monocular 0,286 0,352 0,812 0,417 1,331 0,668 2,679

Model for IC Situations 

Intercept -1,849 0,360 -5,136 0,0001 

Baseline vision 0,00 0,00 

MRV 1,001 0,432 2,314 0,021 2,720 1,188 6,587

Monocular 1,129 0,428 2,637 0,008 3,094 1,368 7,456

SE: standard error; OR: odds ratio; CI: confidence interval.

- Page 14-15 Line 287-297.” For the CC situations, monocularity was the only significant condition in the model (p=.024). Model parameters showed that racing drivers in the monocular condition have 2.1 (95% CI 1.112 - 4.111) greater odds of having a collision than racing drivers in the Baseline binocular condition. For CI situations, the GLMM showed that neither monocularity nor MRV was significant. Finally, for IC situations, both monocularity and MRV were significant (p=.008 and p=.021, respectively). Model parameters showed that recently monocular racing drivers had 3.094 (95% CI 1.368 – 7.456) greater odds of having a collision compared with the baseline condition, and racing drivers with MRV had 2.72 (95% CI 1.188, 6.587) times greater odds of having a collision compared with the baseline visual condition.”

- Page 20 line 392-399. “We conducted a series of GLMMs to determine if monocularity or MRV were predictive of an accident and the odds of a collision occurring, given the specific visual condition. A GLMM was conducted for both the CC and II hazard situations (Table 4). The analysis indicate the absence of significant overdispersion for CC situations model (�2= 590.933; ratio= 0.903; rdf= 654; p= .963) and for II situations model (�2= 242.486; ratio= 0.973; rdf= 320; p= .999).”

- Page 20 line 400-404. “Monocularity in the CC situations model was the only significant condition in the logistical regression models (p<.00015). Model parameters showed that racing drivers in the monocular condition have 6.4934.77 (95% CI 3.004911 -– 11,1337.571) greater odds of having a collision compared with racing drivers in the baseline condition.”

- Page 20-21 line 406-411. 

Table 4. Parameter estimates from the two GLMMs for the 2 driving situations related to the occurrence of collisions for each visual condition

Parameter Estimate SE Z value p OR 95% CI low 95% CI high

Model for CC Situations

Intercept -2,041 0,274 -7,448 0,0001 

Baseline vision 0,00 0,00 

MRV 0,376 0,273 1,377 0,168 1,457 0,851 2,523

Monocular 1,87 0,263 7,102 0,0001 6,493 3,911 11,133

Model for II Situations 

Intercept -1,554 0,348 -4,463 0,0001 

Baseline vision 0,00 0,00 

MRV 0,083 0,351 0,239 0,811 1,087 0,538 2,211

Monocular 0,112 0,360 0,312 0,755 1,119 0,542 2,321

SE: standard error; OR: odds ratio; CI: confidence interval.

- Page 21 line 416-419. Specifically, the results of the two experiments show that, compared with the baseline condition, drivers under the recent monocular condition are 2.1 to 6.54.7 times more likely to collide with target vehicles, depending on the driving situation.

- Page 28 line 604-608. “20. Bates D, Mächler M, Bolker BM, Walker SC. Fitting linear mixed-effects models using lme4. J Stat Softw. American Statistical Association; 2015;67. doi:10.18637/jss.v067.i01

- 21. R Development Core Team. R: A Language and Environment for Statistical Computing. Vienna.: R Foundation for Statistical Computing; 2015.

- 22. Bolker BM, Brooks ME, Clark CJ, Geange SW, Poulsen JR, Stevens MHH, et al. Generalized linear mixed models: a practical guide for ecology and evolution. Trends in Ecology and Evolution. 2009. pp. 127–135. doi:10.1016/j.tree.2008.10.008”

We removed:

Page 9 line 197-202. “We theorized that all situations are different from each other because the angle of the curve, the speed of approach, and the position of the hazardous event are never the same. Furthermore, a single driver could have experienced the same situation twice (test re-test) but in a completely different visual context with a major impact on driving performance. Thus, we considered every situation as a separate sample for the logistic regression.”

2. Also in the first experiment, how many participants were in CC, CI, and IC, respectively? On average, there were six subjects in each scenario. The authors need to justify why the sample size is sufficient to detect meaningful difference. A similar justification may also be needed for the second experiment

We understand the reviewer’s concern:

Each of the 18 participants passed the three conditions CC, CI and IC. 

We have added the following in the manuscript:

- Page 12-13 line 266-271. “We evaluated 18 participants who passed all three visual conditions. In each condition, drivers were confronted with three types of dangerous situations. Six tests were proposed for each dangerous situation. Eliminating missing data related to the failed or incorrectly performed tests, we collected 232 observations for the CC situations, 220 observations for the CI situations and 214 observations for the IC situations.

- Page 20 line 387-391. “We evaluated 31 participants who passed all three visual conditions. In each condition the drivers were confronted with two types of dangerous situations. Eight tests were proposed for the CC situations and four tests for the II situations. If we eliminate the failed or incorrectly performed tests, we collected 658 observations for the CC condition and 325 observations for situation II.”

We added in the statistical analysis paragraph:

- Page 8-9 line182-190. “The minimum number of participants required was determined by an a priori power analysis. Our objective was to compare the occurrence of accidents by visual condition and report the estimated ORs using a logistic regression (the analysis was modified during the revision process). To determine the number of observations required per group (n), we assumed that the accident rate expected in the control condition (Baseline) would be 5% and 20% in the monocular condition with a power of at least 90% and an α of .05. We obtained a number of 100 observations per group, or a minimum number of 17 participants taking into account the number of tests per driving situation.”

3. I wonder whether the authors could explain why the reaction time was not measured and compared in the second experiment?

We appreciate the reviewer’s question. In experiment 1, drivers have to detect and react to a sudden intrusion of an opponent’s racing car into the participant's trajectory in a turn. They have no choice but to provide an action (breaking or turning) to not crash. We can thus determine a reaction time between the intrusion into the visual field and the beginning of an avoidance action. In the second experiment, the opposing cars come from behind and interfere with the drivers in their trajectory around an up coming bend. The main solution used by the drivers to avoid accidents in this experiment was to take a slightly wider turn as soon as they see the opposing car. However, as the opposing car comes from behind the driver, there is no way to determine exactly when the driver sees the opposing car and, thus, no way to determine the delay from the time the driver sees the opposing car to the time he reacts to it. 

4. In the discussion, the authors attempted to explain how monocular vision or MRV impacts driving performance during racing through reaction time. Is it possible that the authors could do some analysis to show the connection between reaction time and collision?

We appreciate the reviewer’s question and have performed correlation analyses between reaction times and accident rates for each visual condition, regardless of the type of situation. As the variables were normally distributed, we used a Pearson correlation. 

Regarding the baseline visual condition, the results of the Pearson correlation indicated that there was no significant association between reaction time and accident rates, (r = .28, p = .26).

Regarding the MRV visual condition, the results of the Pearson correlation indicated that there was a significant positive association between reaction time and accident rates, (r=.64, p=.004).

Regarding the monocular visual condition, the results of the Pearson correlation indicated that there was a significant positive association between reaction time and accident rates, (r=.48, p=.044).

The results of the correlational analysis show that the increase in reaction time has an impact on involvement in accidents only under conditions where vision is impaired. This can be explained by the fact that in baseline conditions, the accident relies more on the intrinsic qualities of each driver to be able to avoid the accident. On the other hand, when vision is impaired, there is a substantial increase in reaction time, which leaves only a small chance for the driver to avoid the accident, given the dynamics of the cars.

We added:

- Page 10 line 206-207. “We analyzed the relationships between reaction times and accident rate for each visual condition, regardless of the type of situation, using Pearson’s correlation."

- Page 16 line 325-334. “Correlations between reaction times and accident rate.

As the variables were normally distributed, we used a Pearson correlation.

Regarding the baseline visual condition, the results of the Pearson correlation indicated that there was no significant association between reaction times and accident rates, (r = .28, p = .26). Regarding the MRV visual condition, the results of the Pearson correlation indicated that there was a significant positive association between reaction times and accident rates, (r=.66, p=.003). Finally, regarding the monocular visual condition, the results of the Pearson correlation indicated that there was a significant positive association between reaction times and accident rates, (r=.48, p=.048).”

- Page 22 line 429-431. “. In addition, correlation analysis allows us to observe that the longer the reaction time increases in this visual condition, the greater the risk of an accident.”

- Page 22- line 442-445. “. Nevertheless, correlation analysis allows us to observe that, for the monocular visual condition, the longer the reaction time increases, the greater the risk of an accident, whereas this is not the case in baseline visual condition.”

---

## [Decision Letter · Decision Letter 1]

21 Oct 2019

PONE-D-19-19649R1

Implications of monocular vision for racing drivers

PLOS ONE

Dear Dr. Adrian,

Thank you for submitting your manuscript to PLOS ONE. After careful consideration, we feel that it has merit but does not fully meet PLOS ONE’s publication criteria as it currently stands. Therefore, we invite you to submit a revised version of the manuscript that addresses the points raised during the review process.

Please address the Reviewer 2's comments in the minor revision. 

We would appreciate receiving your revised manuscript by Dec 05 2019 11:59PM. To enhance the reproducibility of your results, we recommend that if applicable you deposit your laboratory protocols in protocols.io, where a protocol can be assigned its own identifier (DOI) such that it can be cited independently in the future. For instructions see: http://journals.plos.org/plosone/s/submission-guidelines#loc-laboratory-protocols

We look forward to receiving your revised manuscript.

Kind regards,

Zhixia Li

Academic Editor

PLOS ONE

Additional Editor Comments (if provided):

Please address the Reviewer 2's comments in the minor revision.

Reviewers' comments:

Reviewer's Responses to Questions

**Comments to the Author**

1. If the authors have adequately addressed your comments raised in a previous round of review and you feel that this manuscript is now acceptable for publication, you may indicate that here to bypass the “Comments to the Author” section, enter your conflict of interest statement in the “Confidential to Editor” section, and submit your "Accept" recommendation.

Reviewer #1: All comments have been addressed

Reviewer #2: All comments have been addressed

2. Is the manuscript technically sound, and do the data support the conclusions?

Reviewer #1: Yes

Reviewer #2: Yes

3. Has the statistical analysis been performed appropriately and rigorously? 

Reviewer #1: Yes

Reviewer #2: Yes

4. Have the authors made all data underlying the findings in their manuscript fully available?

Reviewer #1: Yes

Reviewer #2: Yes

5. Is the manuscript presented in an intelligible fashion and written in standard English?

Reviewer #1: Yes

Reviewer #2: Yes

6. Review Comments to the Author

Reviewer #1: All my comments have been addressed.

Thanks.

Reviewer #2: (1) It may be worth pointing out that the ANOVA is for the repeated measurements.

(2) The authors found that there is no strong correlation between the reaction time and the accident rate in the baseline condition, and argued that "this can be explained by the fact that in baseline conditions, the accident relies more

on the intrinsic qualities of each driver to be able to avoid the accident." I wonder if the authors can provide any evidence to support their claim.

7. PLOS authors have the option to publish the peer review history of their article (what does this mean?). If published, this will include your full peer review and any attached files.

Reviewer #1: Yes: Hao Liu

Reviewer #2: No

---

## [Author Response · Author response to Decision Letter 1]

7 Nov 2019

Responses to Reviewer #2:

Reviewer #2: (1) It may be worth pointing out that the ANOVA is for the repeated measurements.

Answer: 

We appreciate the reviewer’s question. we have modified:

Page 9 line 196-197 “We also analyzed driver reaction time in crash avoidance using two-way analysis of variances (ANOVAs) with repeated measures.” by “We also analyzed driver reaction time in crash avoidance using two-way repeated measures ANOVA.

Page 13 line 282-283 “The two-way ANOVA with repeated measures showed significant main effects of the visual condition” by “The two-way repeated measures ANOVA showed significant main effects of the visual condition”.

(2) The authors found that there is no strong correlation between the reaction time and the accident rate in the baseline condition, and argued that "this can be explained by the fact that in baseline conditions, the accident relies more on the intrinsic qualities of each driver to be able to avoid the accident." I wonder if the authors can provide any evidence to support their claim.

We appreciate the reviewer’s question. There is no specific evidence we are currently aware of and this claim is only made based on our analysis and observations. We should have used instead :“this may be explained by the fact that in baseline conditions, the accident relies more on the intrinsic qualities of each driver to be able to avoid the accident". If the reviewer is aware of any evidence we would be open to including it here.

---

## [Decision Letter · Decision Letter 2]

25 Nov 2019

Implications of monocular vision for racing drivers

PONE-D-19-19649R2

Dear Dr. Adrian,

We are pleased to inform you that your manuscript has been judged scientifically suitable for publication and will be formally accepted for publication once it complies with all outstanding technical requirements.

With kind regards,

Zhixia Li

Academic Editor

PLOS ONE

Additional Editor Comments (optional):

Reviewers' comments:

Reviewer's Responses to Questions

**Comments to the Author**

1. If the authors have adequately addressed your comments raised in a previous round of review and you feel that this manuscript is now acceptable for publication, you may indicate that here to bypass the “Comments to the Author” section, enter your conflict of interest statement in the “Confidential to Editor” section, and submit your "Accept" recommendation.

Reviewer #2: All comments have been addressed

2. Is the manuscript technically sound, and do the data support the conclusions?

Reviewer #2: Yes

3. Has the statistical analysis been performed appropriately and rigorously? 

Reviewer #2: Yes

4. Have the authors made all data underlying the findings in their manuscript fully available?

Reviewer #2: Yes

5. Is the manuscript presented in an intelligible fashion and written in standard English?

Reviewer #2: Yes

6. Review Comments to the Author

Reviewer #2: The authors have addressed all comments I have in the previous reports. I do not have any further comment.

7. PLOS authors have the option to publish the peer review history of their article (what does this mean?). If published, this will include your full peer review and any attached files.

Reviewer #2: No

---

## [Editor Report · Acceptance letter]

4 Dec 2019

PONE-D-19-19649R2 

Implications of monocular vision for racing drivers 

Dear Dr. Adrian:

I am pleased to inform you that your manuscript has been deemed suitable for publication in PLOS ONE. Congratulations! Your manuscript is now with our production department. 

With kind regards,

on behalf of

Dr. Zhixia Li 

Academic Editor

PLOS ONE